# How Do Children Play with Toy Trains and for What Benefits? A Scoping Review

**Salim Hashmi** 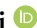

Department of Psychology, Institute of Psychiatry, Psychology and Neuroscience, King's College London, London SE1 1Ul, UK; salim.hashmi@kcl.ac.uk

**Abstract:** Children play with different toys in different ways which may be associated with different developmental outcomes. While existing work has investigated different categories of toys, differences may also be present within specific toy categories. Therefore, understanding how specific toys promote play behaviours and their associated developmental outcomes has important implications for teachers, parents, caregivers, and researchers. To better understand how children play with toy trains, whether groups of children show a particular preference for toy trains and what (if any) associated benefits there are for playing with toy trains, 36 studies published in psychology and educational databases up to December 2022 were reviewed. A key finding emerged regarding the importance of the structured, realistic, and familiar nature of toy trains being important for facilitating pretend play as well as social collaboration behaviours during social play. Whilst findings in relation to gender-stereotyped preferences for playing with toy trains were mixed and no gender differences were found in research investigating play styles, neurodivergent children were found to have a preference for toy trains. These findings are important given that certain play styles, pretend play in particular, have been associated with benefits in children's executive function, language, creativity, and social understanding.

**Keywords:** toys; toy-play; toy trains; train play; play

## 1. Introduction

Play is central to many children's daily lives and children enjoy playing with a variety of games, digital devices, and toys [1,2]. For example, in one survey of 292 four- to twelve-year-old children, over a third of children reported that playing with toys was one of their favourite activities when playing alone [3], though this preference decreases with age [3,4]. In terms of specifically playing with toy vehicles, when 270 seven-year-old children's parents were asked whether their children liked to play with toy figures like trucks, cars, and trains, over half of the parents reported them as doing so, though this was higher in the boys than the girls [5]. Therefore, many children enjoy spending their free time playing with toys and specifically playing with toy vehicles like trains.

When children incorporate toys, such as toy trains, into their play, they often act as 'a prod to the imagination' ([6] p. 24]), in encouraging children's pretend play (the playful distortion of reality to behave in a nonliteral "as if" mode [7]) and prompting them to create storylines, characters, and sound-effects to accompany and structure how they play [8,9]. Indeed, children's behaviours with toys can reflect pretend enactments of roles and actions through their speech and behaviour; children narrate stories to accompany their play with toys; additionally, children manage or negotiate when playing with others the allocation and setting out of toys as well as proposing where one toy may stand for something else [10–13]. Children's play with toys can be expected given the identity, function, and physical properties of the toy [14,15] or can be creative in transforming toys to resemble something different, both of which are positively associated with one another [13]. Pretend play was considered to develop from the age of one or two years, peak at the age of four years, and decline thereafter

until the age of seven years, though it is now recognized that pretend play continues beyond this [1,16].

In addition to pretend play, toys such as toy trains may allow children to engage in construction play, involving building things with materials [1]. This encompasses manipulating toys such as blocks or bricks to build specific things [1,17], but also includes connecting and setting up toys such as train tracks [17,18]. This type of play is common in both pre-school aged children [1] as well as older children (6–7-year-olds) [4]. Indeed, setting up toys as a part of play has been found to be more common than playing pretend with toys in expected or creative ways [18,19] and is negatively associated with them both [13,18].

Therefore, children play with different toys and play with these toys in different ways. Both the toys themselves and play behaviours may be associated with the development of different outcomes. In general, toys that are structured and realistic have been found to prompt more pretend play in some children than toys that are not realistic [20,21], which in turn is associated with the development of aspects of children's cognitive and social development [22]. For example, playing with toys like dolls with others and alone in comparison to playing with tablet games alone has been found to activate the posterior superior temporal sulcus (pSTS), an area of the brain associated with social processing [23]. This pattern of brain activity was explained by toys prompting the reflection of the internal states of the characters and others more in comparison with playing digital games [24,25]. Whereas construction toys like blocks and train tracks prompt non-social, sensorimotor, and construction play in some children [17,26], which in turn is associated with the development of spatial reasoning [27] and reading and maths skills [28].

However, even within toy 'categories', specific types or themes of toys can lead to different ways of playing. For example, when playing with superhero figures compared to more generic toy figures, boys showed more pro-social behaviour, negotiations about pretend play, and sharing meanings with the superhero toys, but showed more variety in pretend themes with the generic toys [29,30]. Therefore, it is important to consider the ways in which specific types of toys might prompt different ways of playing, which in turn may be associated with specific developmental outcomes.

Finally, there are individual and group differences in the toys children play with and how they play with such toys. For example, research indicates consistent gender differences in toy preferences, where children show a preference for gender-matched toys [see [16,31]. Further, some research indicates that boys are more likely to engage in pretend play that depends on transforming objects and toys than girls [32,33], whereas construction play has been found to be no different according to gender [4]. Additionally, there are mixed findings in relation to whether neurodivergent children, particularly autistic children, play in different ways. In regard to preferences for toys, autism has been characterised by restricted interests in toys and activities for some children [34] and different preferences for toys compared to neurotypical children [35]. For play behaviours, where one review found that symbolic pretend play was less evident in autistic children compared to neurotypical peers, particularly in free-play paradigms [36], other research has found that symbolic pretend play is rare and not significantly different between neurotypical and autistic children [35]. Further, some research has found that non-pretend play behaviours are more common in autistic children [35], whereas others have found that only sensorimotor play is more common in autistic children compared to neurotypical children [37]. Therefore, it is also important to understand the ways in which types of toys might prompt different ways of playing in different children, which in turn may be associated with particular developmental outcomes.

*Aims of the Paper*

Children include a variety of different toys in their play which children play with in different ways and are associated with different developmental outcomes. Whilst much of the existing work has investigated different categories of toys more generally (e.g., [21,26]),

differences may also be present within particular toy categories [29,30]. Understanding more with regard to the impact of specific toys has important implications, for example, in informing teachers, parents, and caregivers which toys are made available for children to play with. Therefore, the present review focuses specifically on children's play with toy trains given that such toys may prompt pretend play given their realistic nature [20,21], however the accompanying accessories such as train tracks may provide children the opportunity to engage in construction play as well [17]. Toy trains were focused on specifically, rather than toy vehicles in general, as findings from previous research revealed value in focusing on a particular category of a toy (e.g., superhero dolls vs. dolls in general [29,30] as well as toy trains often being accompanied by tracks and accessories that would promote a variety of play types, including construction play, unlike other types of toy vehicles such as planes or cars [17]. Given that the aim of this review was to identify and map out the existing literature regarding children's play with toy trains, as opposed to establishing the quality and consistency of the evidence, a scoping review was chosen over a systematic review [38]. Therefore, the existing literature was synthesised and reviewed in regard to: (1) the different ways in which children play with toy trains; (2) how playing with toy trains might be associated with development; and (3) whether some children have more of a preference for playing with toy trains than others.

## 2. Materials and Methods

**Search.** A systematic search of articles was carried out on December 1st 2022 following PRISMA guidelines [39] and based on the strategy of a previous systematic review in the area of play with fathers [40]. Four databases covering psychological and educational research areas were searched: PsycInfo, PubMed, ERIC, and the British Education Index. Search terms related to play used in Amodia-Bidakowska and colleagues' [40] review were included, though the strategy itself differed. Key word searches encompassing 'play', 'pretend play', 'games', 'toys', and 'recreation' were used to search all fields, which was identical in each database searched. This was combined with relevant database specific subject-headings, which differed slightly between databases due to the availability of these subject-headings within the databases (see Appendix A for search strategies employed for each database). Search terms related to train play included 'trains', 'train sets', and two specific brands that were present on the website of a popular UK toy store (ToysRUs): 'Thomas the Tank Engine' and 'Hornby'. Finally, the search was limited to extract studies exploring child or adolescent samples to reflect the research question. Additional limits related to the type of article returned were not included to ensure that 'grey literature' was also included. The searches in these databases returned a total of 1768 articles, from which 124 were duplicates, resulting in a total of 1644 articles for screening (see Figure 1).

The titles and abstracts of these articles were initially screened, and articles that were any additional duplicates (*n* = 3) or articles that were not relevant to the present review as they used the terms 'train' or 'play' in a different way to the intended search (e.g., 'train' used as 'teach'/'learn'/'practice'; 'play' used in the context of sports or 'role play' in a therapeutic or educational context) or did not reflect a study in which children's play or use of a toy train in another task was explored were excluded (*n* = 1518). A conservative approach was adopted to ensure that papers were not prematurely excluded; for example, papers that mentioned a play session or game but not specifically a train toy were included, as were papers that referenced a train toy but not explicitly a play session. This process resulted in a total of 123 articles for the full-text assessment of eligibility.

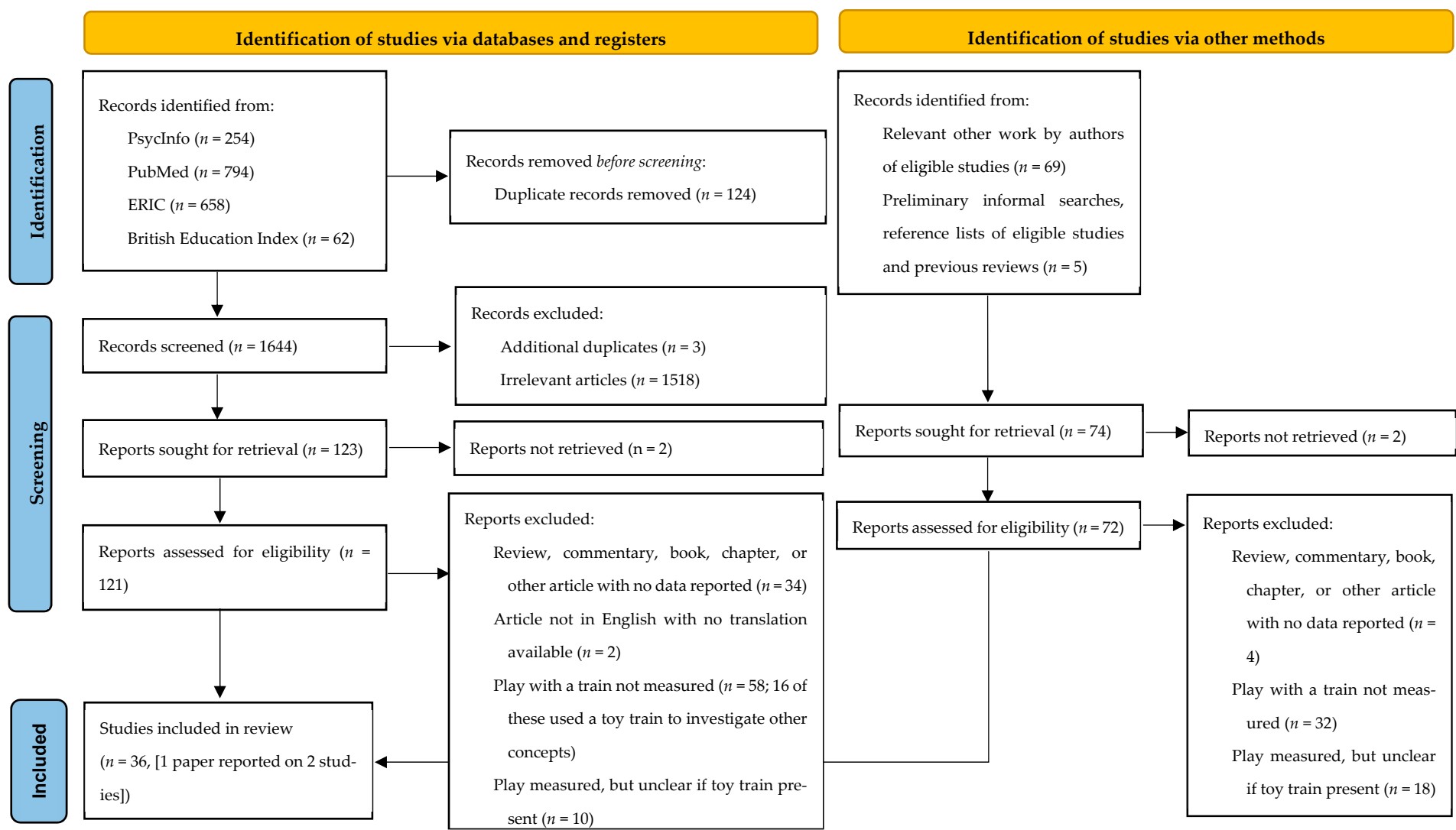

**Figure 1.** PRISMA flow diagram [39] depicting records identified, excluded, and included in the final review.

The full texts of these articles were assessed using the following inclusion criteria: (1) research contributing new data or re-analysis of previously collected data; (2) assessed play with a toy train using any relevant method (e.g., questionnaire, observation, etc.); and (3) includes children and adolescents up to 18 years of age. No exclusion criteria were set in relation to the population sampled (e.g., neurodivergent and neurotypical children were included) or whether playing with trains was investigated in relation to additional variables; these characteristics of the studies were extracted later. Articles that used toy trains to explore other constructs (perception of speed, imitation, memory, reasoning, object permanence) were excluded, unless children were given the opportunity to freely play with the train at some point. Previous relevant reviews, commentaries, reports, books, or book chapters that did not report new data were excluded from the review, but were read and the references sections assessed for any additional relevant articles (see below). This process resulted in a total of 17 articles from database searching for data extraction and review (see Figure 1).

Additional articles were identified through searching the reference lists of the 17 papers considered eligible for review; through searching other published works by the authors of the 17 papers considered eligible for review (see Appendix for details as to the process for doing this); and through searching the reference lists of the reviews, commentaries, books, book chapters, and other articles which were excluded as they did not report new data. This resulted in an additional 18 papers that met the criteria and were included in the review (see Figure 1).

**Data extraction.** Key details of each of the 36 (1 paper summarised the results of 2 relevant studies) studies were extracted and presented in Appendix B. Information was extracted from each article on: (1) the sample that was investigated; (2) the duration, frequency, and type of play session that was measured or included in the study; (3) the variables with which playing with trains were associated with; and (4) the results pertaining to the children's play with trains.

These articles were then categorised according to: (1) whether the sample was a community sample or included a clinical group (either as a comparison group or the population of interest); (2) the general outcome measure(s) of interest (social behaviour, play behaviour, social cognition [including language], or toy preference); (3) the experimental design or type of study; and (4) whether the analysis explored the outcome measure specifically for the toy train. A narrative synthesis of this research is presented below, identifying the details and trends in the research and findings as well as gaps in the current literature. Due to the variety of study designs and outcome measures in the research, neither a meta-analysis or formal risk of bias assessment was conducted. Detailed information about the studies is presented in Appendix B.

## 3. Results

### 3.1. General Description of Studies

Over half of the studies (57%) sampled children and their families in North America (18 in US, 3 in Canada), and a quarter of the studies (25%) investigated children in Europe (3 in UK, 4 in France, 1 in Italy, 1 in Finland). The remaining 17% of the studies investigated children's play with trains in Australia, Turkey, China, and Qatar. Three-quarters of the studies recruited both boys and girls, with no known clinical diagnoses, from different socio-demographic backgrounds, and from different ethnic, cultural, or racial groups. The other 25% of studies looked at how neurodivergent children or children with different long-term health conditions played with trains. On average, the children that were a part of the research studies were 5 years old, though this ranged from 1-year-old to 15-year-old children.

In terms of the research questions investigated in the 36 studies, half explored the different ways in which children play with toy trains (i.e., engagement in pretend play, construction play, social vs. solitary play). Over two-thirds of the studies involved children playing together with another person, of which 31% looked at how children socialised with other people when playing with toy trains. A few studies (19%) investigated how playing

with toy trains was associated with aspects of children's development, in particular their language, social behaviours, and their social understanding. Finally, over a third of studies (39%) explored children's preferences for toy trains compared to other toys and compared to other groups of children (e.g., according to neurodivergence or gender).

### 3.2. How Do Children Play with Toy Trains?

Studies involving children from China [41], Australia [42], and America [18,43] found that children engage in pretend play with toy trains. Boys and girls in these studies were generally similar in their pretend play with the toy trains [18,42] and it was to the same extent as other toys [42]. However, although the amount of pretend play with toy trains was the same, one study found that the themes played out within their pretence might be different; when playing with trains, pretend play was often centred around what the toys were (i.e., they were playing 'train themes') compared to other types of toys where the themes of play were quite different from the themes of the toys [18,44]. Finally, studies have also found that children can learn to pretend play with toys like trains by watching another person playing pretend [45].

However, some research also found that in some situations children showed less pretend play with toy trains. One study found that neurodivergent children were less likely to play pretend with the toy trains [42], and children who were less familiar with toy trains also showed less pretend play with these toys [46]. In the studies, children instead played with the toy trains by setting up the tracks, trains, and other related toys (e.g., the train station) [18]. Similar to pretend play, children can learn to play setting up and constructing toy trains by watching another person play with them by constructing the trains and tracks [47]. Finally, some research investigated how children produced humour when playing with toys, including toy trains [48,49].

### 3.3. Children's Social Play with Toy Trains

Some of the studies found that how children communicated with each other when playing with toys was different depending on who they were playing with [50–53] as well as with what they were playing with [18,43]. Specifically, when playing with toy trains, children were found to be more social (being both more pro-social and disruptive) compared to when they were playing with other toys [43] and were having conversations that reflected their own goals, desires, and intentions [18]. Studies also found that when children were with their peers, younger children play with toys like trains on their own but with others as they grow older [54,55]. Children's social play and interactions were also facilitated by the familiarity of the toy, resulting in more spontaneous and fluid interactions when playing with other children [56].

### 3.4. How Is Children's Play with Toy Trains Associated with Their Development?

Limited studies looked at children's development specifically. One study found that children's language production increased over 3 months when children were given toys, including trains, to play with [57]. However, the children who did not play with trains also produced more language after 3 months, so this may have reflected children's language developing with age. Some research also found that children talked about the minds and internal states of others (people, toys, and characters they have created) when playing with toys, including toy trains [18,58].

### 3.5. Children's Preferences for Toy Trains

In studies that looked at how different groups of children enjoy playing with different toys, neurodivergent children have been found to particularly enjoy playing with toy trains compared to other toys and more so than neurotypical children [35,42,54,56]. In addition, some research found that boys prefer playing with toy trains more than girls [59–61]. However, gender-stereotyped toy preferences were not consistently found in the research [9,42], and no gender differences were found in regards to the outcomes in the research discussed

above. Finally, one of the studies looked at whether play was different in American children from different cultural and sociodemographic backgrounds and found that it was no different [9].

## 4. Discussion

In the present scoping review, the existing literature was systematically searched in order to identify and map out the research regarding children's play with toy trains. Thirty-six studies were reviewed in order to explore the different ways in which children play with toy trains, whether some children have more of a preference for playing with toy trains than others, and how playing with toy trains might be associated with development. Very few studies directly investigated whether playing with toy trains was directly associated with developmental outcomes compared to other toys, and limited studies directly compared how the same child played with toy trains versus other toys, highlighting these areas as gaps in this literature. Rather, studies investigated characteristics of children's play with toys, which included toy train sets, and explored associations of these characteristics with respect to developmental outcomes. Therefore, in this section, the outcomes of the review are discussed in relation to their implications for children's development.

From the studies reviewed, children sampled from North America, Australasia, and Asia as well as children from different sociodemographic backgrounds, all engaged in pretend play with toy trains. When engaging in this form of play, children played out themes or scenarios that were expected and reflected what the toys are, for example, moving the train along the track and 'walking' toy passengers onto the train to be picked up and taken to their destinations [18]. Familiarity with what the toys are therefore may be important for facilitating pretend play, and indeed children evidenced more pretence with a toy train compared to a toy tractor as the toy train reflected something that children are familiar within their environment and that are realistic, giving them a reference point for how they should play with these toys [41]. This is in line with the existing work highlighting that children demonstrate more pretend play with toys that are structured and realistic compared to toys that are not [20,21].

These findings are noteworthy as pretend play in general is associated with gains in several domains. For example, children who pretend play more have improved executive function, creativity, and imagination [62–65], with evidence supporting this direction of effects [66,67]. Further, pretend play has been associated with children's language development [22], perspective taking [68,69] and empathy [70], though the direction of effects for these outcomes are less clear. Therefore, given that toys that are familiar and realistic, such as toy trains, may allow children to engage in pretend play more than less familiar and unrealistic toys [20,21], providing an opportunity for children to play with such toys may have benefits for their development.

Additional support for the importance of the familiar, realistic, and structured component of toy trains comes from the research focusing on children's speech when playing. When playing with toy trains with a partner, children's conversations were more pro-social and disruptive, collaborative, and reflected on children's goals for the play as compared to when playing with more 'open-ended' toys such as a farm-set [18,43]. This was likely a result of the 'close-ended' and structured nature of the train set indicating to children clearly their use, in turn requiring less discussion and clarification on the themes and uses of the toy before being able to play with them [43]. Further, children's social play with toy trains additionally included humour [48,49] and references to the internal states of other people and the toys and characters [18,58], both of which are associated with children's developing social understanding [19,71].

In line with the existing literature, children also engaged in construction play with toy trains by constructing and setting up the tracks, trains, and related accessories [18,47]. This finding is important, as construction play has also been found to positively impact the development of children's spatial reasoning [27] and reading and maths skills [28]. Further, it has been found that the relationship between construction play and maths abilities might

be mediated by aspects of executive function and specifically visuospatial memory [72]. Therefore, in addition to such toys having the potential to provide developmental benefits through allowing children to engage in pretend play, gains in some domains may also be present through toys, such as toy trains, allowing for construction play.

Finally, the studies reviewed indicated mixed findings as to whether some children have more of a preference for toy trains compared to other toys and other children. Findings across studies were consistent in that neurodivergent, particularly autistic children, enjoyed playing with toy trains more so than other toys and compared to neurotypical children, in line with the existing research regarding general toy preferences in neurodivergent children being different [35]. Further, in one of the studies in neurodivergent children, pretend play was less common in free-play tasks in this group compared to neurotypical peers [42]. This finding also aligns with the existing literature in that pretend play is seen less in neurodivergent children when assessed in free-play contexts [36]. However, findings related to gender-stereotyped preferences for toy trains were mixed, and no gender differences were found in relation to the ways in which children played with the toys or behaved while doing so. Therefore, whilst it may be that there are differences in the toys boys and girls would choose to play with, this is unlikely to translate to differences in how they would play with them and the associated outcomes if given the opportunity to.

This review has some limitations. First, although a thorough search strategy was devised to result in a comprehensive collection of relevant papers for review, it is likely that some studies that included toy trains as part of a selection of toys for play, but were not specifically mentioned in the papers, will have been missed. Second, though we endeavoured to ensure that the search strategy was identical between each database, and this was the case for the key-word searches, there were subtle differences present in subject headings searched in each database (e.g., 'Dramatic Play' was only available in ERIC). However, our key-word searches within each database always included 'play', ensuring that papers related to other types of play, such as construction or dramatic play, in databases that do not contain these subject headings would have still been included in the review. Third, not all studies reported analyses according to the toy trains when included as part of a wider assortment of toys, and therefore, some findings may not be specific to toy trains compared to other toys. Fourth, the review only included accessible studies published in the English language or with translations available, therefore there is a potential that publication biases impact the conclusions made. Finally, it is important to note that associations found between playing with trains and developmental outcomes may not be specific to toy trains. Indeed, other toys with similar properties (e.g., toy cars with roads or tracks) may elicit similar patterns of playing, which in turn may also be associated with these outcomes. Further, child characteristics, additional environmental influences, and other aspects of children's play with toys (e.g., the length of time playing) might impact the associations between children's play with toy trains and developmental outcomes.

## 5. Conclusions and Implications

This review of the existing literature on children's play with toy trains indicated that children from different backgrounds play with toy trains socially as well as on their own, and by including pretence and humour, as well as by engaging in construction play, setting up, and organising the toys. The structured, realistic, and familiar nature of such toys was a key feature that facilitated children's play with toy trains in these different ways. These findings have implications for teachers, parents, and all caregivers when considering the types of toys to make available to children in order to support their play activities and the associated developmental outcomes. In particular, an important implication is that the same toy, in this case, toy trains and the tracks and accessories accompanying them, can prompt different ways of playing, but these play patterns can also be modelled or scaffolded to children [45,47], though this should not be done to encourage children to play in a particular way. These findings may also influence researchers' decision-making when including toys in their research, in so far as demonstrating how different toys may promote

different patterns of behaviour, which may be of relevance depending on the research question of interest. Future research therefore could further investigate the nuances in how children play with specific toys and their associated benefits as well as more detailed comparisons between toys.

**Funding:** This research was funded by Mattel UK Ltd. (Slough, UK).

**Institutional Review Board Statement:** Not applicable.

**Informed Consent Statement:** Not applicable.

**Data Availability Statement:** Not applicable.

**Acknowledgments:** Not applicable.

**Conflicts of Interest:** Funding for this project was provided by Mattel UK Ltd. Mattel funded personnel costs for this project but did not play a direct role in the collection or analysis of the data; interpretation of the results; or writing of the manuscript.

## Appendix A. Complete Search Strategy
**PsycInfo**

| Search | Key Word (KW) or Subject Heading (SH) | Terms | Results |
|---|---|---|---|
| #1 | KW | Play OR preten * OR game* OR toy OR recreation (all fields) | 230,463 |
| #2 | SH | Recreation OR doll play OR toys OR childhood play behaviour OR games OR childhood play development OR playfulness | 27,222 |
| #3 | KW | Train OR trains OR train set OR train track OR train play OR Thomas the tank engine OR Thomas and friends OR Hornby (all fields) | 18,855 |
| #5 | SH | Railroad trains | 659 |
| #6 | | 1 OR 2 | 230,744 |
| #7 | | 3 OR 5 | 18,855 |
| #8 | | 6 AND 7 | 1215 |
| #9 | | Limit 8 to childhood (birth to 12 years) or adolescence (13 to 17 years) | 254 |

*Note*. For key words related to 'Train', the term 'train *' was not used as it resulted in many irrel-evant articles, largely related to the term 'training'. * reflects truncation of the search term.

**PubMed**

| Search | Key Word (KW) or Subject Heading (SH) | Terms | Results |
|---|---|---|---|
| #1 | KW | "play" [all fields] OR "preten*" [all fields] OR "game*" [all fields] OR "toy" [all fields] OR "recreation" [all fields] | 902,704 |
| #2 | SH/MeSH | "recreation" [MeSH terms] OR "play and playthings" [MeSH terms] | 239,869 |
| #3 | KW | "train" [all fields] OR "trains" [all fields] OR "train set" [all fields] OR "train track" [all fields] OR "train play" [all fields] OR "Hornby" [all fields] | 59,758 |
| #4 | | 1 OR 2 | 1,103,474 |
| #5 | | 4 AND 3 | 1215 |
| #6 | | 5 Filters: child birth–18 years | 794 |

*Note.* For key words related to 'Train', the term 'train*' was not used as it resulted in many irrele-vant articles, largely related to the term 'training'. Subject headings/MeSH terms in PubMed were different to those used in PsycInfo due to database differences, and an analogous term for 'railroad trains' did not exist. * reflects truncation of the search term.

**ERIC**

| Search | Key Word (KW) or Subject Heading (SH) | Terms | Results |
|---|---|---|---|
| #1 | KW | Play OR preten* OR game* OR toy OR recreation (all fields) | 97,451 |
| #2 | SH | Recreation OR toys OR dramatic play OR games | 17,256 |
| #3 | KW | Train OR trains OR train set OR train track OR train play OR Thomas the tank engine OR Thomas and friends OR Hornby (all fields) | 10,099 |
| #4 | | 1 OR 2 | 97,451 |
| #5 | | 4 AND 3 | 658 |

*Note.* For key words related to 'Train', the term 'train*' was not used as it resulted in many irrele-vant articles, largely related to the term 'training'. Subject headings in ERIC were different to those used in PsycInfo due to database differences, and an analogous term for 'railroad trains' did not exist. Additionally, there is no function-ality to set a limit to childhood given the focus of the database on education. * reflects truncation of the search term.

**British Education Index**

| Search | Key Word (KW) or Subject Heading (SH) | Terms | Results |
|---|---|---|---|
| #1 | KW | Play OR preten* OR game* OR toy OR recreation (all fields) | 13,928 |
| #2 | SH | Play OR imaginative play OR symbolic play OR games | 1925 |
| #3 | KW | Train OR trains OR train set OR train track OR train play OR Thomas the tank engine OR Thomas and friends OR Hornby (all fields) | 974 |
| #4 | | 1 OR 2 | 13,928 |
| #5 | | 4 AND 3 | 62 |

*Note.* For key words related to 'Train', the term 'train*' was not used as it resulted in many irrele-vant articles, largely related to the term 'training'. Subject headings in the British Education In-dex were different to those used in PsycInfo due to database differences, and an analogous term for 'railroad trains' did not exist. Additionally, though there was functionality to set a limit to childhood; this removed many relevant papers and so was not used.

In addition to these searches, the reference list of the first author of the eligible studies were screened and reviewed for inclusion. In practice, this was done by searching for the authors' records in PsycInfo and Web of Science and exporting those with relevant titles to Endnote to identify duplicates and complete abstract screening. This resulted in 584 articles, and when duplicates were removed by screening, it resulted in 69 papers which underwent the full-text assessment of eligibility. Of these, 13 papers satisfied the inclusion criteria and were included in the review.

**Appendix B. Summary of Studies Included in the Review**

**Table A1.** A review of the literature published between 1900 and 2022 on train play in children and adolescents up to 18 years of age.

| Author(s) | Sample Information | Play Task, Measure, and Other Variables | Results in Relation to Train Play |
|---|---|---|---|
| Abuhatoum et al. (2020) [73] | *n* = 46<br>Age range: T1 focal child = 56.4 months (4.7 years); T2 focal child = 94.58 months (7.9 years)<br>Gender: Mixed<br>Children played with older and younger siblings; mixed gender compositions as well as with friends<br>Country: US | Children freely played with either a train, farm, or village set for 15 min at 2 time points.<br>Conflict, power resources, and power effectiveness coded. | No significant differences in measures/variables according to play set, focal child gender, or gender composition. |

**Table A1.** *Cont.*

| Author(s) | Sample Information | Play Task, Measure, and Other Variables | Results in Relation to Train Play |
|---|---|---|---|
| Alhaddad et al. (2019) [56] | *n* = 10 Age range: 7–10 years old Gender: All boys Sample with ASD diagnosis Design: Country: Qatar | Four 6 min long experiments video-recorded using different stimuli from the list below. Experiment 1 was a free play session to explore children's preference for the first 5 toys below. Experiment 2 and 3 were to explore preferences for social humanoid robots and Thomas the Tank Engine trains, respectively. Experiment 4 was to explore if bubbles generated from a toy train increase its appeal. Play materials:<br>- Rubber ball<br>- Cymbals<br>- Colourful plastic train (not Thomas the Tank Engine)<br>- Small humanoid robot<br>- Wooden truck with a carrier<br>- Two interactive social robots<br>- Three Thomas the Tank Engine trains<br><br>Behaviours analysed included the duration of the experiment; the duration of interactions with the experimenter within the experiment; preference for toy (indicated by speech, longest interaction duration, or most preferred); or an unclear preference. | In experiment 1, the plastic train was the 4th most preferred toy (out of 5). "Three [children] liked the features of the train, such as the colours and wheels". In experiment 3, just over 46% of children showed interest to the Thomas the Tank Engine trains with 20% showing interest in the other train. "Children were more excited in this experiment". Familiarity with the train played a role in making interactions fluid and spontaneous, with children re-enacting crashing scenes and producing sound effects of trains. In experiment 4, nearly all children (93.3%) preferred the train with bubbles over their previously most favoured train. The duration of interaction increased in terms of interacting with the bubbles. |
| Besler & Kurt (2016) [47] | *n* = 3 Age range: 5–6 years old Gender: All boys Sample with ASD diagnosis Country: Turkey | Mothers trained and guided to produce a video modelling how to build a Lego train as a play skill. Children were then shown the video, and the children were assessed on whether they learned the play skill and maintained this. | Children were better able to build the Lego train after successive sessions of watching the training videos and one week after the last session. |
| Chu et al. (2006) [41] | *n* = 140 Age range: 13–59 months Gender: 82 boys, 58 girls; 72 children with developmental delay Country: China | Symbolic play test (SPT; validated in Western cultures) was modified to be appropriate to a Chinese sample. In the SPT, children are presented with 4 different 'situations' of increasing difficulty in which an experimenter places toys in front of the child who is encouraged to play with them. Children's behaviour is then scored according to how they play with the toy and whether it demonstrates some form of pretend play. In the modified version used in the present study, a tractor and trailer were replaced with a train and train track and cutlery was replaced with chopsticks to be more appropriate for Chinese children. | Children performed better (evidenced symbolic/pretend play) more when the 'situation' was modified to include a train instead of a tractor. |
| Cordoni et al. (2016) [74] | *n* = 129 Age range: 3–5 years old (mean ages 38.75–63.07 months) Gender: 73 boys, 56 girls Country: Italy | Recordings made for 6 h a day, for 69 days in a Kindergarten which had toys available, including a train set. From the recordings, children's behaviour (aggression, affiliation, play) was coded. | No distinct analyses by toy set or of train set. No gender differences in children's aggression or who the target of aggression was; in groups of boys, there was more of a reliance on physical contacts than girls. Gender segregation in play was seen only in boys, regardless of age. |

**Table A1.** *Cont.*

| Author(s) | Sample Information | Play Task, Measure, and Other Variables | Results in Relation to Train Play |
|---|---|---|---|
| Dalgin-Eyiip & Ulke-Kurkcuogluu (2021) [45] | *n* = 4<br>Age range: 5–8 years old<br>Gender: 3 boys, 1 girl<br>Sample with ASD diagnosis<br>Country: Turkey | Videos were made whereby 'peers' modelled three pretend play skills (tea time play, hairdressing play, and railway train play).<br>These videos were shown to the children who were reinforced with treats for watching it, and whether they engaged in the pretend play modelled was assessed. | Children's pretend play skills/replication of the modelled actions increased following sessions where the videos were shown. |
| DeLoache et al. (2007) [75] | *n* = 177<br>Age range: 11 months–6 years old (mean 35.1 months)<br>Gender: 84 boys, 93 girls<br>Country: US | Parents completed questionnaires and interviews in relation to their children's 'extremely intense interests'. | In total, 116 children were identified as having or having had one intense interest. These interests were more likely to be present in boys. Five of these children (all boys) had an extreme interest in trains. |
| Desha et al. (2003) [42] | *n* = 24<br>Age range: 3.42–7.17 years old (*M* = 5.21 years old)<br>Gender: 17 boys, 7 girls<br>Sample with ASD diagnosis<br>Country: Australia | Children videotaped for 15 min of unstructured play and 15 min of structured play.<br><br>- Play materials:<br>- Thomas the Tank Engine train set<br>- Gross motor toys (Thomas the Tank Engine ride-on, Thomas punching bag)<br>- Construction toys (blocks, craft materials, etc.)<br>- Infant toys (activity board etc.)<br>- Dress-up accessories (handbag, crown, etc.)<br>- Dolls<br>- House toys (plastic food, tea set)<br>- Action figures<br>- Plastic animals<br>- Dress-up clothes<br><br>Play behaviour and play object choices coded. Behaviour categorised as not attending; unrelated behaviour; labelling; giving/showing; attempting to terminate session; exploration; sensorimotor play; relational play; functional play; and symbolic play. | Of the play objects, the Thomas the Tank Engine train set, gross motor, and construction play objects were played with the most, more than expected by chance.<br>No gender differences found in how often children played with the train set (or any other toys other than the dolls). |
| Dominguez et al. (2006) [35] | *n* = 59<br>Children with diagnosis of ASD: *n* = 24; mean age = 5.42 years old; 17 boys, 7 girls<br>Neurotypical children: *n* = 34, mean age = 4.58 years old; 16 boys, 18 girls.<br>Country: Australia | The study formed part of a larger study that included Desha et al. [42] above.<br>Methods were identical to Desha et al. [42] above, with the focus being a comparison between the ASD and neurotypical groups. | Children in the ASD group showed more exploratory (moving or turning it over in hands), sensorimotor (behaviours which do not take into account function of toy [e.g., banging, swinging]), and relational play (playing with two or more objects which does not take into account function of toy [e.g., piling objects, putting objects in a box]) than the neurotypical group. The difference in relation to sensorimotor play was present when looking specifically at the Thomas the Tank Engine train set. Children in the ASD group showed more of a preference for the Thomas the Tank Engine train set, gross motor toys, infant toys, dress-up accessories, actional figures, and plastic animals compared to the neurotypical group. |

**Table A1.** *Cont.*

| Author(s) | Sample Information | Play Task, Measure, and Other Variables | Results in Relation to Train Play |
|---|---|---|---|
| Hobson et al. (2005) [76] | *n* = 32<br>10 mothers with borderline personality disorder (BPD) and 22 mothers with no history of psychiatric conditions<br>Infant age: 47–58 weeks (mean = 54 weeks<br>Infant gender: 16 boys, 16 girls<br>Country: Unstated, likely UK | Mothers and their infants played together with a plastic toy train for 2 min. Maternal relatedness/sensitivity coded based on this interaction. | Mothers with BPD were more 'intrusive insensitive' (how much the mother's actions cut across, took over or disrupted the infant's activities) when playing with the train than mothers without BPD. |
| Howe et al. (2022) [43] | *n* = 44<br>Age: 7 years old (*M* = 7.88; SD = 0.94 years old)<br>Children played with older and younger siblings; mixed gender compositions as well as with friends of the same age and gender.<br>Country: US | Children videotaped in the home either with sibling or friend (counterbalanced) 1 week apart. Children free-played for up to 10 min. Play materials:<br>Children were given a wooden village and train set (including tracks, trains, bridge, crane, people, boats), but whether this was with the sibling or friend was counterbalanced, as was the order of presentation. Conversational turns and shared meaning strategies were coded from videos of the free play. Shared meaning strategies included introductions to play; simple maintenance strategies; semantic tying strategies; clarifications; responses to negotiation; prosocial behaviour; and disruptive behaviour. | When playing with siblings, children used more simple strategies (descriptions and imitations) and clarifications (agreement of ideas and sharing references) with the village set and more prosocial strategies (teaching/helping, social statements, shared affect) with the train set.<br>When playing with friends, children used more simple strategies with the village set and used more introductions (suggesting play themes or calling attention), prosocial behaviour, and disruptive strategies (directives/control statements, negative behaviours, irrelevant behaviours to play) with the train set.<br>No effects of gender composition or birth order were present in relation to playing with the train set. |
| Howe et al. (1993) [46] | *n* = 100<br>Age range: 2.5–5 years old<br>Gender: 60 boys, 40 girls<br>Country: Unstated, likely Canada | All children played in one room organised into activity centres which were designed by early education students. These included: (1) a hospital office; (2) a bakery; (3) a pharmacy; (4) a pirate ship; (5) a pizzeria; (6) an airplane; (7) an animal hospital; (8) a train station; (9) a store; (10) a farm.<br>Children's play in these centres was categorised according to cognitive play (functional, constructive, dramatic, rule-based, exploratory); its social context (solitary, parallel, group); and other non-play behaviour. | Train themed 'center' was less familiar and less likely to elicit dramatic play. |

| Author(s) | Sample Information | Play Task, Measure, and Other Variables | Results in Relation to Train Play |
|---|---|---|---|
| Howe et al. (2022b) [18] | *n* = 52<br>Age: 7 years old (*M* = 7.82; SD = 0.89 years old)<br>Children played with older and younger siblings; mixed gender compositions as well as with friends of the same age and gender.<br>Country: US | Children videotaped in the home either with sibling or friend (counterbalanced) 1 week apart. Children free-played for up to 15 min. Play materials:<br>Children were given a wooden village and train set (including tracks, trains, bridge, crane, people, boats), but whether this was with the sibling or friend was counterbalanced, as was the order of presentation. Conversational turns were coded from videos as well as play scenarios, object use, and internal state language. Play scenarios were coded as set-up/organisation; expected; or creative. Object use was coded as set-up/organisation; expected use/transformation; creative use/transformation; or no objects used. Internal state language coded as references to cognitions; goals; emotions; and preferences. | In general, children engaged in set-up/organisation scenarios more than expected scenarios, and expected scenarios more than creative ones. In relation to the toy sets, children engaged in expected scenarios more with the train set than with the village set. In general, children used objects to set-up/organise the most and transformed them in creative ways the least. In relation to the toy sets, children engaged in expected object use more with the train set than with the village set. In general, children referred to goals more so than cognitions, followed by emotions and preferences which were not significantly different. In relation to the toy sets, children referred to goals more when playing with the train set than when playing with the village set.<br>No effects of gender composition or birth order were present in relation to playing with the train set. |
| Lamminmäki et al. (2012) [59] | *n* = 47<br>Age: 14 months<br>Gender: 21 boys, 26 girls<br>Country: Finland | Toy preference test from 9 toys that were considered as female-preferred (a tea set, a soft doll, a baby doll with a bathtub), male-preferred (a truck, a train, and a parking toy with motorbikes), or neutral (a teddy bear, a soft picture book, and a set of keys). Children played freely with the toys for 8–10 min. Time during which child played with each toy was calculated. | Boys played more with the train compared to the girls; girls played more with the baby doll than the boys. No other gender difference was present for the time spent with other toys.<br>Testosterone levels positively associated with playing with the train in the girls, but not the boys. |
| Le Maner-Idrissi (1996; Experiment 1) [60] | *n* = 24<br>Age: 24 months<br>Gender: 12 boys, 12 girls<br>All oldest child or only child<br>Country: Unstated, likely France | Two children of the same gender were brought into one room for 20 min and presented with stereotypically male toys (a train, a pistol, and a workbench); female toys (a baby doll, a vanity, and a tea set); and neutral toys (a phone, a ball, and a farm). Children's choice of objects and imitative behaviour was recorded. | Children's toy preferences were sex-stereotyped. |
| Le Maner-Idrissi (1996; Experiment 2) [60] | *n* = 24<br>Age: 24 months<br>Gender: 12 boys, 12 girls<br>All oldest child or only child<br>Country: Unstated, likely France | Two children of mixed gender were brought into one room for 20 min and presented with stereotypically male toys (a train, a pistol, and a workbench); female toys (a baby doll, a vanity, and a tea set); and neutral toys (a phone, a ball, and a farm). Children's choice of objects and imitative behaviour was recorded. | Girls in the presence of a boy preferred the female toys to male and neutral toys, but boys' preferences depended on the composition of the dyad–when with boys they preferred the male toys, but in the presence of a girl, they no longer exhibited their preferences. Boys also chose significantly more female toys when with another girl than another boy (based on experiments 1 & 2). |

**Table A1.** *Cont.*

| Author(s) | Sample Information | Play Task, Measure, and Other Variables | Results in Relation to Train Play |
|---|---|---|---|
| Le Maner-Idrissi (1996; Experiment 3) [60] | *n* = 24<br>Age: 24 months<br>Gender: 12 boys, 12 girls<br>All oldest child or only child<br>Country: Unstated, likely France | Two children of the same gender were brought into the room and presented with male toys (a robot, a pistol, a garage, a helicopter, a jeep and its trailer, 4 small cars, a workbench, a construction game, and a train) first and then a week later only the female toys (a doll, a baby basket, a stroller, a baby doll with a bottle, 4 pieces of jewellery, a vanity, a tea set, and a market stand). Children's choice of objects and imitative behaviour was recorded. | Girls chose significantly more female toys than male toys when they were presented separately, but the boys did not show a preference. |
| Le Maner-Idrissi & Renault (2006) [61] | *n* = 48<br>Age range: 34–52 months<br>Gender: 24 boys, 24 girls<br>All children knew each other<br>Country: Unstated, likely France | Two children of mixed gender were brought into one room for 20 min and presented with stereotypically male toys (a train, a pistol, and a workbench); female toys (a baby doll, a vanity, and a tea set); and neutral toys (a phone, a ball, and a farm). Children's choice of objects and interactions (solitary, parallel, or interactive play) was recorded. | Three-year-old boys prefer male toys to either the female or neutral ones, but three-year-old girls did not prefer the male toys over female ones.<br>Four-year-old boys preferred male and neutral toys over the female ones, and four-year-old girls preferred male toys over female toys. |
| Leach et al. (2015) [58] | *n* = 65<br>Age: 56.4 months<br>Children played with older and younger siblings; mixed gender compositions as well as with friends of the same age and gender.<br>Country: US | Children videotaped in the home either with sibling or friend (counterbalanced) 1 week apart. Children free-played for up to 15 min. Play materials:<br>At time 1, children were given either a wooden village, farm, or train set (including tracks, trains, bridge, crane, people, boats), but whether this was with the sibling or friend was counterbalanced, as was the order of presentation. At time 2, children were given either the village set or farm set but whether this was with the sibling or friend was counterbalanced, as was the order of presentation.<br>Play sessions were coded for the number of conversational turns and internal state language. | No analyses by play set.<br>Children referred to cognitions more at time 2 than time 1, specifically with their siblings, and were more likely to refer to shared internal states at time 2. Children talked about shared goals and their own cognitions more at time 2 than time 1, and children talked about emotions about the toys more at time 1 than time 2. At time 1, children with an older sibling talked more about goals and cognitions than children playing with a younger sibling. |
| Leach et al. (2019) [50] | Time 1 *n* = 65<br>Time 2 *n* = 46<br>Time 1 age = 56 months<br>Time 2 age = 94.58 months<br>Children played with older and younger siblings; mixed gender compositions as well as with friends of the same age and gender.<br>Country: US | Children videotaped in the home either with sibling or friend (counterbalanced) 1 week apart. Children free-played for up to 15 min. Play materials:<br>At time 1, children were given either a wooden village, farm, or train set (including tracks, trains, bridge, crane, people, boats), but whether this was with the sibling or friend was counterbalanced, as was the order of presentation. At time 2, children were given either the village set or farm set but whether this was with the sibling or friend was counterbalanced, as was the order of presentation.<br>Play sessions were coded for the number of conversational turns and the ways in which children constructed shared meanings in their play | No analyses by play set.<br>Children used more positive shared-meaning strategies with friends compared to siblings and more introductions with siblings than friends. Children used more simple strategies, building of ideas, and prosocial strategies at time 1 compared to time 2. Specifically, when children were using a play voice, children used more simple strategies at time 2 and more clarifications at time 1. |

**Table A1.** *Cont.*

| Author(s) | Sample Information | Play Task, Measure, and Other Variables | Results in Relation to Train Play |
|---|---|---|---|
| Leach et al. (2019) [51] | Time 1 *n* = 44<br>Time 2 *n* = 46<br>Time 1 age = 56.4 months<br>Time 2 age = 96.77 months<br>Children played with older and younger siblings; mixed gender compositions as well as with friends of the same age and gender. | Children videotaped in the home either with sibling or friend (counterbalanced) 1 week apart. Children free-played for up to 15 min. Play materials:<br>At time 1, children were given either a wooden village, farm, or train set (including tracks, trains, bridge, crane, people, boats), but whether this was with the sibling or friend was counterbalanced, as was the order of presentation. At time 2, children were given either the village set or farm set but whether this was with the sibling or friend was counterbalanced, as was the order of presentation.<br>Play sessions were coded for the number of conversational turns, the 'connectedness' of children's conversations, the quality of interactions, and the emotional tone of the children's conversations. | No differences found in outcome measures in relation to the play set, therefore this was not analysed separately.<br>Children were more cooperative at time 2, and children were more cooperative with friends than with siblings. Children engaged in long sequences of connectedness with friends than with siblings and were more likely to engage in short sequences with siblings than with friends. Children showed a more positive tone with friends compared to siblings, and a more negative tone with siblings than with friends. |
| Leach et al. (2022) [52] | Time 1 *n* = 65<br>Time 2 *n* = 46<br>Time 1 age = 56 months<br>Time 2 age = 94.58 months<br>Children played with older and younger siblings; mixed gender compositions as well as with friends of the same age and gender.<br>Country: US | Children videotaped in the home either with sibling or friend (counterbalanced) 1 week apart. Children free-played for up to 15 min. Play materials:<br>At time 1, children were given either a wooden village, farm, or train set (including tracks, trains, bridge, crane, people, boats), but whether this was with the sibling or friend was counterbalanced, as was the order of presentation. At time 2, children were given either the village set or farm set but whether this was with the sibling or friend was counterbalanced, as was the order of presentation.<br>Play sessions were coded for the number of conversational turns and the 'connectedness' of children's conversations. | No differences found in outcome measures in relation to the play set, therefore this was not analysed separately.<br>Children made more failed attempts at establishing connectedness and engaged in more self-talk when playing with siblings than friends, and they maintained connectedness more with friends than siblings. At time 1, children ended connected interactions more often than their siblings, and siblings engaged in more self-talk and unclear statements than at time 2 only. The balance of participation did not differ between children and friends at either time 1 or time 2. |
| Leach et al. (2022) [53] | Time 1 *n* = 65<br>Time 2 *n* = 46<br>Time 1 age = 56 months<br>Time 2 age = 94.58 months<br>Children played with older and younger siblings; mixed gender compositions as well as with friends of the same age and gender.<br>Country: US | Children videotaped in the home either with sibling or friend (counterbalanced) 1 week apart. Children free-played for up to 15 min. Play materials:<br>At time 1, children were given either a wooden village, farm, or train set (including tracks, trains, bridge, crane, people, boats), but whether this was with the sibling or friend was counterbalanced, as was the order of presentation. At time 2, children were given either the village set or farm set but whether this was with the sibling or friend was counterbalanced, as was the order of presentation.<br>Play sessions were coded for the number of conversational turns, the strategies used to create shared meanings, and the 'connectedness' of children's conversations. | No analyses by play set.<br>Children used simple strategies most often to initiate and sustain connectedness.<br>Children engaged in prosocial behaviour and used the play voice when initiating connectedness with their friends more than with siblings. Children used clarifications when sustaining connectedness more often with siblings than friends. |

**Table A1.** *Cont.*

| Author(s) | Sample Information | Play Task, Measure, and Other Variables | Results in Relation to Train Play |
|---|---|---|---|
| Murphy et al. (1986) [77] | *n* = 20<br>Age: 14.5 years old<br>Gender: 13 boys, 7 girls.<br>All children with developmental delays<br>Country: Unstated, likely UK | Three 5 min observations of children playing with a panda, a car, and a train separately. Children then experienced control and experimental sessions (randomised and counterbalanced). Both conditions consisted of 5 sessions with each toy and children were shown how the toy works, and in the experimental condition, children were encouraged to interact with it.<br>Children's contact with the toys measured. | Children showed more contact with the car and the train in the experimental condition, but not for the panda.<br>Modelling of play behaviour facilitated children's play. |
| Neisworth et al. (2002) [78] | *n* = 4<br>Age: 3–6 years old<br>Gender: All boys<br>Sample with diagnosis of ASD<br>Country: Unstated, likely US | Videos were created of the children demonstrating 'spontaneous requesting' (asking for an object, action, or help) after being trained to do so, from a 30 min play session in the home. Children chose what toys to play with (which included a Thomas the Tank Engine toy set).<br>Children then watched these videos once a day for 5 days, and were observed in the school setting for the target behaviour of spontaneous helping. | Children's spontaneous requesting increased from baseline after the intervention, and was maintained afterwards. |
| O'Bleness (2016) [57] | *n* = 155<br>Time 1 age = 30.68 months<br>Time 2 age = 33.16 months<br>Gender: 75 girls, 80 boys<br>74 mother–child dyads were in the play-as-usual group and 81 mother–child dyads were in the experimental group.<br>Country: US | Children and their mothers played together for 10 min at time 1 and time 2 with an assortment of toys, including a train set.<br>In between these sessions, all dyads completed eight play sessions (4 at home, 4 in the lab), once a week for 20 min. The train set was included during some of these.<br>In addition, the two groups completed a training session: for the control group, they were asked to play with their child as they usually would on a daily basis; for the experimental group, mothers were instructed to learn and engage in the child's game (child-led play that was rewarded).<br>Children's language production coded from the play sessions at time 1 and time 2. | No analyses by play set.<br>Children and mothers' language associated from time 1 to time 2. Mothers' language production decreased from time 1 to time 2, but children's language production increased from time 1 to time 2 (and was particularly marked for children with the lowest language production at time 1). |
| Paine et al. (2021) [48] | Time 1 *n* = 65<br>Time 2 *n* = 46<br>Time 1 age = 56 months<br>Time 2 age = 94.58 months<br>Children played with older and younger siblings; mixed gender compositions as well as with friends of the same age and gender.<br>Country: US | Children videotaped in the home either with sibling or friend (counterbalanced) 1 week apart.<br>Children free-played for up to 15 min.<br>Play materials:<br>At time 1, children were given either a wooden village, farm, or train set (including tracks, trains, bridge, crane, people, boats), but whether this was with the sibling or friend was counterbalanced, as was the order of presentation. At time 2, children were given either the village set or farm set but whether this was with the sibling or friend was counterbalanced, as was the order of presentation.<br>Play sessions were coded for the presence of humour in children's play. | No analyses by play set.<br>Humour did not differ according to relationship (friends vs. siblings) or from time 1 to time 2. Children's production of humour with sibling at time 1 was associated with humour production with a friend, both at time 1 and time 2. Children playing with an older sibling produced more humour with their older sibling than children playing with their younger sibling. |

**Table A1.** *Cont.*

| Author(s) | Sample Information | Play Task, Measure, and Other Variables | Results in Relation to Train Play |
|---|---|---|---|
| Paine et al. (2019) [49] | *n* = 86<br>Age: 7.82 years old<br>Children played with older or younger siblings; 31 mixed gender compositions and 55 same gender compositions.<br>Country: US | Children videotaped in the home with sibling or friend free-playing for 15 min.<br>Play materials:<br>Children either played with village set (*n* = 42) or train set (*n* = 44).<br>Play sessions were coded for conversational turns and the presence of humour in children's play. | More humour produced when playing with the village toys compared to the train set.<br>The production of humour was dependent on the other sibling/partner in the play session. Humour differed according to the age and gender composition of the children.) In general, the boys produced more humour than the girls. |
| Parten (1933) [54] | *n* = 34<br>Age: Pre-school age<br>Country: US | Children observed daily for one minute each during morning free play in pre-school.<br>Records made of the play activity and the number and characteristics of children in each group. | Of the 11 most popular activities reported, playing with train toys was the 3rd most popular. There were no clear age-related trends for train play, but it was more popular with boys. Younger children tended to play with the trains on their own, whereas for older children, train play was a social activity involving building the tracks or stations with blocks. |
| Petrakos & Howe (1996) [44] | *n* = 31<br>Age range: 43–64 months<br>Gender: 18 boys, 13 girls<br>Country: Unstated, likely Canada | Children were semi-randomly assigned to groups of four but matched on dramatic play abilities and peer familiarity.<br>Each group entered the dramatic play centre in groups of four and played for 10 min in each dramatic play centre. For the intervention, four centres were provided: (1) extended housekeeping; (2) train station which were set up to promote either solitary or group interactions.<br>Children's play categorised according to type of play (functional, constructive, dramatic) and degree of sociality (solitary, parallel, group, onlooker, unoccupied). | The social/solitary designs of the centres promoted the social/solitary type of play, respectively. Children engaged in role play that was theme-related and consistent with the theme of the centre. |
| Ritter-Brinton & Beattie (1994) [79] | *n* = 1<br>Age: 5 years old<br>Gender: Not stated<br>Deaf child of hearing parents<br>Country: Unstated, likely Canada | Eight 15 min play sessions, alternating between a train set and a doctor role play kit. Focal child played with two same-aged male peers who were also deaf.<br>Different types of play behaviour were recorded: initiation, maintenance, shift, and termination. | No explicit analyses of differences between toys, but data are presented separately. However, children showed more play behaviours for the doctor role play kit than the train set. Further, the child showed similar play behaviours for all categories other than termination (ending a play sequence), which were less for the train set as compared to the doctor role play kit. |
| Roggman (1989) [55] | *n* = 108<br>Age: 36 10-month-olds; 36 15-month-olds; 36 29-month-olds<br>Gender: 10-month-old group = 13 boys, 23 girls; 15-month-old group = 17 boys and 19 girls; 29-month-old group = 13 boys and 23 girls<br>Country: Unstated, likely US | Children played with two toy trains—one that was non-social in that it could be played with alone, and the other was social in that it required adult help (an object needed to move the train was out of reach). Both trains were presented to each child, but the position (on the left or right) was counterbalanced.<br>Toy preference was measured according to the time spent looking at each train. | The youngest and oldest group did not show a preference for either toy, but the 15-month-olds showed a preference for the 'social' train compared to the 'non-social' train. |

**Table A1.** *Cont.*

| Author(s) | Sample Information | Play Task, Measure, and Other Variables | Results in Relation to Train Play |
|---|---|---|---|
| Spektor-Levy et al. (2017) [80] | *n* = 106<br>Age range: 5–10 years old<br>Gender: 52 girls, 54 boys.<br>Country: UK | Children completed problem-solving tasks that involved building a train track to match a pre-set shape or building a LEGO model according to a pre-set plan.<br>Children's private (self-directed) speech and private (self-directed) gestures coded. | Children's self-directed speech positively associated with age for both tasks. Aspects of children's self-directed speech were correlated across tasks, but only for the boys. |
| Tavassoli et al. (2020) [81] | Time 1 *n* = 63<br>Time 2 *n* = 44<br>Time 1 age = 4.56 years old; Time 2 age = 8.06 years old<br>Children played with older and younger siblings, mixed gender compositions as well as with friends of the same age and gender.<br>Country: US | Children videotaped in the home either with sibling or friend (counterbalanced) 1 week apart. Children free played for up to 15 min. Play materials:<br>At time 1, children were given either a wooden village, farm, or train set (including tracks, trains, bridge, crane, people, boats), but whether this was with the sibling or friend was counterbalanced, as was the order of presentation. At time 2, children were given either the village set or farm set but whether this was with the sibling or friend was counterbalanced, as was the order of presentation.<br>Play sessions were coded for how children responded to their play partner's request for help. | No analyses by play set.<br>Children were more likely to refuse to be prosocial with siblings compared to friends. The ways in which children refused to be prosocial differed according to age (i.e., from time 1 to time 2) and according to the request being made. |
| Trawick-Smith et al. (2015) [9] | *n* = 60<br>Age: 48.8 months<br>Gender: 32 girls, 28 boys<br>Country: US | Nine toys were placed in the classroom for a 20 min video recorded free play. The play of children who chose to play with the toy was recorded.<br>Toys included: wooden train set; Bristle blocks; Duplo bricks; Lincoln logs; Measure up! cups; Rainbow people; Castle bucket set; Shape, model and mould; tree blocks.<br>Play assessed in terms of 'quality'—whether the play evidenced: (1) thinking and learning; (2) problem solving; (3) curiosity and inquiry; (4) sustained interest; (5) creative expression; (6) symbolic transformation; (7) interactions with peers; (8) autonomous play. | Train set not amongst the highest for play quality, but also not the lowest. No differences according to gender, children from low SES backgrounds showed higher 'play quality' with trains compared to other toys, but then no difference in quality between those from higher or lower SES backgrounds. |

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
