# Peer review of "How Do Children Play with Toy Trains and for What Benefits? A Scoping Review"

_ejihpe, doi:10.3390/ejihpe13100149_

Round 1
Reviewer 1 Report
The paper is well-written, but I have three main concerns that require clarification and/or addressing.
The first concerns the search for benefits of a specific toy for children’s development. Assessing the developmental benefits of a specific toy in research is a challenging task unless an intervention is conducted. Firstly, child development is a complex interplay of various domains, making it difficult to pinpoint how a single toy impacts a specific development aspect. Secondly, the uniqueness of each child, with distinct personalities and developmental paths, complicates generalizing the benefits of a toy. Additionally, many developmental advantages may only emerge after extended exposure or interactions with other toys and experiences. These factors collectively contribute to the challenge of conclusively demonstrating the developmental benefits of a particular toy in research. Thus, the results only indicated a limited number of studies. With this in mind, perhaps the title of the review and the main focus can be changed. Many other angles explored bring value to literature, such as children’s toy preferences.
My second primary concern pertains to the inconsistencies in the terminology used across various databases. If there was a particular reason for the differences in search terms, that needs to be explained in the article.
As my research team and I are nearing the completion of a systematic review, one of our key criteria was maintaining consistency in our search terms in various databases. When limitations were imposed, we remained consistent. I understand the differences in keyword search options in various databases. However, we encountered a notable discrepancy in this article. For example, while one database was searched using the term "dramatic play," others were explored using "imaginative play," "symbolic play," and "preten*."It's essential to recognize that these play types possess subtle nuances, and it would be beneficial if the article's introduction included operational definitions for these various play types. Did the author consider encompassing all these nuances under a broader conceptual umbrella?
Lastly, the article appears to need a comprehensive explanation of the significance of trains within the context of constructive play. The author explained in the introduction that trains primarily facilitate pretend and constructive play (with accessories). Then, the constructive play component vanished from the explanations and analysis. Constructive play, as a category, encompasses activities where children create, build, or manipulate objects to construct something new. Trains, often considered an integral element of constructive play, can serve as a prime example of this category.
The absence of a clear and detailed explanation regarding the importance of trains in constructive play leaves the audience with an incomplete understanding. However, the article's deficiency in clarifying the role of trains in constructive play has a cascading effect on the choice of search terms. Thus, the article doesn't provide a robust framework for understanding the role of trains in the context of constructive play. My suggestion would be to include a more thorough discussion on this aspect.
Here are some minor suggestions for improvement:
1. In the introduction, two key elements need addressing:
(i) Why the focus on trains specifically? It's important to articulate the significance of trains in the context of children's play. Explaining why trains were chosen over other toys like cars, for example, would strengthen the argument for the review's relevance. While significant figures like trains are mentioned, it's crucial to delve into the rationale behind this selection.
(ii) Consider discussing age differences in children's interactions with toys and open-ended materials. Although Trawick-Smith's work in 2015 explored age differences, including this information would provide valuable insights.
2. Given the author's assertion that trains primarily facilitate pretend play and constructive play (with accessories), the introduction would greatly benefit the reader if definitions of these two play types were included. This would enhance understanding and alignment with the author's arguments.
3. In the aims section (lines 97-100), a font size inconsistency needs correction.
4. Regarding Figure 1, adjustments are needed in the layout and box sizes to ensure it fits the page with the line numbers. Currently, the figure becomes illegible after line 143, which hinders comprehension.
5. Line 183 references this article as systematic, which may require clarification or correction based on the article's title; on that note, if the author could explain why they chose a scoping review instead of a systematic review, it would benefit the audience.
Author Response
Dear editorial team,
Please find attached our revised manuscript, “How Do Children Play with Toy Trains and for what Benefits? A Scoping Review.” I thank the reviewers for their positive and constructive feedback and have made substantial revisions to the manuscript, which are described below. Our changes are highlighted in the manuscript, and the page/line numbers I refer to below correspond to the revised version submitted. The numbers ascribed to the reviewer’s comments I have added to aide in making these revisions.
Responses to Reviewer 1:
The paper is well-written, but I have three main concerns that require clarification and/or addressing.
- The first concerns the search for benefits of a specific toy for children’s development. Assessing the developmental benefits of a specific toy in research is a challenging task unless an intervention is conducted. Firstly, child development is a complex interplay of various domains, making it difficult to pinpoint how a single toy impacts a specific development aspect. Secondly, the uniqueness of each child, with distinct personalities and developmental paths, complicates generalizing the benefits of a toy. Additionally, many developmental advantages may only emerge after extended exposure or interactions with other toys and experiences. These factors collectively contribute to the challenge of conclusively demonstrating the developmental benefits of a particular toy in research. Thus, the results only indicated a limited number of studies. With this in mind, perhaps the title of the review and the main focus can be changed. Many other angles explored bring value to literature, such as children’s toy preferences.
Thank you for this detailed comment and for providing a space to further reflect on this within the manuscript. In the first draft, I attempted to articulate the rationale for investigating a specific toy category (toy trains) as analogous work has been done with other types of toys (e.g., dolls) and specific categories within this toy type (e.g., superhero dolls). I have detailed this further with particular reference to toy trains in response to comment 4 below.
With regard to the focus on developmental benefits, in light of the comments of other reviewers (e.g., “the author did a tremendous job in noting the importance of different play types with a common toy in early childhood settings”) I have not at this stage changed the title of the review or the main focus. Instead, I have reflected on your important points in the discussion (lines 394-400) to better frame these findings and acknowledge this concern.
- My second primary concern pertains to the inconsistencies in the terminology used across various databases. If there was a particular reason for the differences in search terms, that needs to be explained in the article.
As my research team and I are nearing the completion of a systematic review, one of our key criteria was maintaining consistency in our search terms in various databases. When limitations were imposed, we remained consistent. I understand the differences in keyword search options in various databases. However, we encountered a notable discrepancy in this article. For example, while one database was searched using the term "dramatic play," others were explored using "imaginative play," "symbolic play," and "preten*."It's essential to recognize that these play types possess subtle nuances, and it would be beneficial if the article's introduction included operational definitions for these various play types. Did the author consider encompassing all these nuances under a broader conceptual umbrella?
Thank you raising this concern. In the supplementary materials, we did detail the search in each database and note that the key-word searches were identical in each database. There were differences in the subject heading searches due to differences in the availability of particular subject headings within each database. This information has now been moved into the method section (lines 135-138). Further, we have reflected on this in the discussion section on lines 383-389, and state here that our strategy of including ‘Play’ as a keyword search in each database would mitigate that some databases did not have particular subject headings. With this strategy, we are confident that the impact of any inconsistency in the subject heading searches was accounted for by the consistency and specific terms used for the key word searches. The suggestion to include definitions in the introduction for these important terms has been addressed in relation to comment 6 below.
- Lastly, the article appears to need a comprehensive explanation of the significance of trains within the context of constructive play. The author explained in the introduction that trains primarily facilitate pretend and constructive play (with accessories). Then, the constructive play component vanished from the explanations and analysis. Constructive play, as a category, encompasses activities where children create, build, or manipulate objects to construct something new. Trains, often considered an integral element of constructive play, can serve as a prime example of this category.
The absence of a clear and detailed explanation regarding the importance of trains in constructive play leaves the audience with an incomplete understanding. However, the article's deficiency in clarifying the role of trains in constructive play has a cascading effect on the choice of search terms. Thus, the article doesn't provide a robust framework for understanding the role of trains in the context of constructive play. My suggestion would be to include a more thorough discussion on this aspect.
Thank you for highlighting the lack of attention paid to construction play within the original manuscript. The introduction now includes a paragraph summarising this type of play, which includes playing with train tracks etc. (lines 51-59). More specific reference to this type of play and it’s connection to toy trains has been included in other areas of the introduction also when discussing potential developmental outcomes (lines 71-73) and potential gender differences (lines 87-88). Finally, we have returned to the finding in relation to construction play in the discussion and contextualised this in relation to the literature in the introduction (lines 355-364). As noted in response to comment 2 and in the discussion, the inclusion of ‘play’ as a key word search in all databases served to ensure all play patterns, including construction play, were included.
Here are some minor suggestions for improvement. In the introduction, two key elements need addressing:
- Why the focus on trains specifically? It's important to articulate the significance of trains in the context of children's play. Explaining why trains were chosen over other toys like cars, for example, would strengthen the argument for the review's relevance. While significant figures like trains are mentioned, it's crucial to delve into the rationale behind this selection.
Thank you for this suggestion. Changes made to the introduction in order to address comment 3 have highlighted the significance of toy trains. Additionally, in lines 114-120 of the introduction there is now a specific rationale included for focusing on toy trains.
- Consider discussing age differences in children's interactions with toys and open-ended materials. Although Trawick-Smith's work in 2015 explored age differences, including this information would provide valuable insights.
Thank you for this suggestion. Age-related trends in pretend play and construction play have been included on lines 47-50 and 55-56 respectively.
- Given the author's assertion that trains primarily facilitate pretend play and constructive play (with accessories), the introduction would greatly benefit the reader if definitions of these two play types were included. This would enhance understanding and alignment with the author's arguments.
Thank you for this suggestion. A definition of pretend play has now been included in lines 37-38 and a definition of construction play has now been included in the section related to this play type lines 51-55.
- In the aims section (lines 97-100), a font size inconsistency needs correction.
Thank you for highlighting this issue, this has now been corrected.
- Regarding Figure 1, adjustments are needed in the layout and box sizes to ensure it fits the page with the line numbers. Currently, the figure becomes illegible after line 143, which hinders comprehension.
Thank you for highlighting this issue with Figure 1, this has now been altered to fit within the page and line numbers.
- Line 183 references this article as systematic, which may require clarification or correction based on the article's title; on that note, if the author could explain why they chose a scoping review instead of a systematic review, it would benefit the audience.
In the revised manuscript, this line is 194 and the word ‘systematic’ has been removed. An explanation and rationale as to why a scoping review was chosen (as opposed to a systematic review) has been included in the aims of the paper section on lines 120-123.
I thank the reviewers sincerely for their time and helpful comments. Thank you very much for considering this revised manuscript for publication in European Journal of Investigation in Health, Psychology and Education.
Yours sincerely,
Reviewer 2 Report
Thank you very much for the opportunity to review your paper.
This was a highly interesting paper which the author did a tremendous job in noting the importance of different play types with a common toy in early childhood settings. It was useful to see how children’s special needs/additional needs had different preferences and approaches to play. I was particularly interested to read the implication section. Although the author has made significant contributions for this section, I see potential for further discussions. For example, for the author to delve further into noting the implication for teacher pedagogy, implications for practice, as well as for early learning curriculum. The types of resources (toys) that should be guided by what type of practice adopted by educators (e.g. scaffolding) could also be noted.
I thoroughly enjoyed reading the author’s contributions. It was novel and the significance was made aware.
Author Response
Dear editorial team,
Please find attached our revised manuscript, “How Do Children Play with Toy Trains and for what Benefits? A Scoping Review.” I thank the reviewers for their positive and constructive feedback and have made substantial revisions to the manuscript, which are described below. Our changes are highlighted in the manuscript, and the page/line numbers I refer to below correspond to the revised version submitted. The numbers ascribed to the reviewer’s comments I have added to aide in making these revisions.
Reviewer 2:
Thank you very much for the opportunity to review your paper.
This was a highly interesting paper which the author did a tremendous job in noting the importance of different play types with a common toy in early childhood settings. It was useful to see how children’s special needs/additional needs had different preferences and approaches to play. I was particularly interested to read the implication section.
- Although the author has made significant contributions for this section, I see potential for further discussions. For example, for the author to delve further into noting the implication for teacher pedagogy, implications for practice, as well as for early learning curriculum. The types of resources (toys) that should be guided by what type of practice adopted by educators (e.g. scaffolding) could also be noted.
Thank you for this positive review and this suggestion. Further specific discussion of the implications have been included in lines 409-416. Due to the content added to the manuscript to address other reviewer comments, at this stage I have kept this brief so as not to overwhelm a reader.
I thoroughly enjoyed reading the author’s contributions. It was novel and the significance was made aware.
I thank the reviewers sincerely for their time and helpful comments. Thank you very much for considering this revised manuscript for publication in European Journal of Investigation in Health, Psychology and Education.
Yours sincerely,
Reviewer 3 Report
1. The figure should be adjusted within a page.
2. Could you explain why you made a scoping review rather than a systematic review? and could you describe pros and cons of the scoping review for your topic/subject/title?
3. Based on the scoping review, did the manuscript reach the goal of scoping review, such as map the extent, range, and nature of the literature, as well as to determine possible gaps in the literature on a topic? Could you write more to discuss the question on your topic?
Author Response
Dear editorial team,
Please find attached our revised manuscript, “How Do Children Play with Toy Trains and for what Benefits? A Scoping Review.” I thank the reviewers for their positive and constructive feedback and have made substantial revisions to the manuscript, which are described below. Changes are highlighted in the manuscript, and the page/line numbers I refer to below correspond to the revised version submitted. The numbers ascribed to the reviewer’s comments I have added to aide in making these revisions.
Reviewer 3:
- The figure should be adjusted within a page.
Thank you for highlighting this issue with Figure 1, this has now been altered to fit within the page and line numbers.
- Could you explain why you made a scoping review rather than a systematic review? and could you describe pros and cons of the scoping review for your topic/subject/title?
An explanation and rationale as to why a scoping review was chosen (as opposed to a systematic review) has been included in the aims of the paper section on lines 120-123.
- Based on the scoping review, did the manuscript reach the goal of scoping review, such as map the extent, range, and nature of the literature, as well as to determine possible gaps in the literature on a topic? Could you write more to discuss the question on your topic?
Thank you for this suggestion. The discussion has been expanded with specific consideration to the aims of a scoping review in lines 305-307 and 311-313; as well as in terms of implications in lines 409-416.
I thank the reviewers sincerely for their time and helpful comments. Thank you very much for considering this revised manuscript for publication in European Journal of Investigation in Health, Psychology and Education.
Yours sincerely,